# Initial Imaging of Pregnant Patients in the Trauma Bay—Discussion and Review of Presentations at a Level-1 Trauma Centre

**DOI:** 10.3390/diagnostics14030276

**Published:** 2024-01-26

**Authors:** Roisin MacDermott, Ferco H. Berger, Andrea Phillips, Jason A. Robins, Michael E. O’Keeffe, Rawan Abu Mughli, David B. MacLean, Grace Liu, Heather Heipel, Avery B. Nathens, Sadia Raheez Qamar

**Affiliations:** 1Department of Medical Imaging, Sunnybrook Health Science Centre, University of Toronto, Toronto, ON M4N 3M5, Canada; roisin.macdermott@sunnybrook.ca (R.M.); ferco.berger@sunnybrook.ca (F.H.B.); jason.robins@sunnybrook.ca (J.A.R.); michaelezra.okeeffe@sunnybrook.ca (M.E.O.); rawan.abumughli@sunnybrook.ca (R.A.M.); 2Tory Trauma Program, Sunnybrook Health Sciences Centre, University of Toronto, Toronto, ON M4N 3M5, Canada; andrea.phillips@sunnybrook.ca; 3Department of Anesthesia, Sunnybrook Health Sciences Centre, University of Toronto, Toronto, ON M4N 3M5, Canada; davidb.maclean@sunnybrook.ca; 4Department of Obstetrics and Gynecology, Sunnybrook Health Sciences Centre, University of Toronto, Toronto, ON M4N 3M5, Canada; grace.liu@sunnybrook.ca; 5Department of Medicine (Emergency Medicine), Sunnybrook Health Sciences Centre, University of Toronto, Toronto, ON M4N 3M5, Canada; heather.heipel@sunnybrook.ca; 6Tory Trauma Program, Department of Surgery, Sunnybrook Health Sciences Centre, University of Toronto, Toronto, ON M4N 3M5, Canada; avery.nathens@sunnybrook.ca

**Keywords:** trauma imaging, pregnancy, patient safety, radiation concerns

## Abstract

Trauma is the leading non-obstetric cause of maternal and fetal mortality and affects an estimated 5–7% of all pregnancies. Pregnant women, thankfully, are a small subset of patients presenting in the trauma bay, but they do have distinctive physiologic and anatomic changes. These increase the risk of certain traumatic injuries, and the gravid uterus can both be the primary site of injury and mask other injuries. The primary focus of the initial management of the pregnant trauma patient should be that of maternal stabilization and treatment since it directly affects the fetal outcome. Diagnostic imaging plays a pivotal role in initial traumatic injury assessment and should not deviate from normal routine in the pregnant patient. Radiographs and focused assessment with sonography in the trauma bay will direct the use of contrast-enhanced computed tomography (CT), which remains the cornerstone to evaluate the potential presence of further management-altering injuries. A thorough understanding of its risks and benefits is paramount, especially in the pregnant patient. However, like any other trauma patient, if evaluation for injury with CT is indicated, it should not be denied to a pregnant trauma patient due to fear of radiation exposure.

## 1. Introduction

Trauma is the leading non-obstetric cause of maternal and fetal mortality and affects an estimated 5–7% of all pregnancies [1,2]. According to Deshpande et al., motor vehicle collisions (MVCs) are the most common cause of trauma in pregnant women and account for 58.1% of trauma mechanisms, followed by falls in 16.7% and assaults in 14.9% of trauma mechanisms [3,4]. The reported incidence of trauma in pregnancy is lower than the actual incidence due to unreported fatal and non-fatal trauma, thus limiting recognition of its actual magnitude [1]. Injury severity in pregnancy can be subdivided using the Injury Severity Score, a score that is commonly used in trauma and can be calculated after all injuries are known. The scoring system is based on anatomic regions and their regional injury severity, with scores ranging from 1 (lowest) to 75 (highest, not survivable) [5]. The ISS correlates linearly with mortality, morbidity, hospital stay, and other measures of severity, with major injury usually classified as an ISS of 16 or higher. However, due to added complexities in pregnant trauma patients, a subdivision of severity of trauma has been published at the cut-off of an ISS of 9, with traumatic injury severity deemed minor if <9 and major if ≥9 [5]. While fetal loss happens in 50% of major traumas and 1–5% of the minor ones, due to the higher incidence of minor trauma in pregnancy, the majority of documented fetal mortality occurs in minor trauma. Whether in minor or major trauma, fetal outcome is dependent on maternal stabilization, which therefore should be the primary focus of initial management. It is worth noting that in a retrospective analysis of pregnant trauma patients presenting over a 4-year period, Bochicchio et al. found that 11% of pregnancies were incidentally diagnosed, emphasizing the need to treat all female trauma patients of childbearing age as pregnant until proven otherwise [6].

If, like in all other trauma patients, initial evaluation of the pregnant trauma patient in the trauma bay with CT is justified, there should be no hesitation to proceed as per normal routine. A CT scan should not be denied due to fear of radiation exposure, since the risk of missing a potentially life-threatening injury in the mother outweighs any potential risk to the fetus. Being well informed and up to date with regard to the risks and benefits of imaging modalities using ionizing radiation will aid in proper patient management and counseling as needed. This article will first detail this information and subsequently provide an overview of imaging findings per imaging modality as well as summarize key data from our centre’s experience over the past 21 years.

## 2. Radiation Dose Considerations

Concern over fetal exposure to ionizing radiation and the related stochastic teratogenic and carcinogenic effects have dominated the discussions about the use of ionizing radiation in pregnant patients for decades, especially with regard to CT. Technologic developments, such as better detectors, faster scanners with reduced exposure times, automated dose modulation, and iterative reconstruction models that reduce image noise, have resulted in a marked reduction in radiation exposure to patients [7,8]. Knowledge about the deterministic and potential stochastic effects of the ionizing radiation procedures have also advanced, which has resulted, for example, in altered recommendations for shielding [9,10]. Understanding these effects, which are further detailed below, can alleviate anxiety and confusion about the safety of imaging in pregnant trauma patients and will promote effective patient counselling regarding risks and benefits.

All ionizing radiation procedures should adhere to the “as low as reasonably achievable” (ALARA) principle to minimize possible deterministic and stochastic effects of radiation exposure. Deterministic effects relate to risks of a single radiation exposure manifesting when a threshold is exceeded, whereas stochastic effects relate to an accumulating risk with multiple exposures over the course of an individual’s life. The deterministic effects of radiation exposure to a fetus depend on the fetal gestational age and the absorbed dose. The potential biological effects of exposing the developing fetus to radiation in-utero are based on both animal and human models, with the deterministic effects including prenatal death, intra-uterine growth restriction, and congenital malformations, amongst others.

The first trimester is the period of highest risk from exposure, as this is the time of rapid cell proliferation and organogenesis. In the first two weeks after conception, the main risk is failure of blastocyst implantation and therefore failure of pregnancy, seen with the estimated exposure threshold ranging between 50 and 100 mGy [8]. During weeks 3–8 after conception, i.e., the period of organogenesis, the embryo is most vulnerable to radiation-induced congenital malformations, occurring at an estimated exposure threshold of >200 mGy. From weeks 8–15 after conception, the fetus has the highest risk of radiation-induced loss of intelligence quotient (IQ) score and intellectual disabilities if exposed to more than 100 mGy. The stochastic fetal radiation effects are, however, expressed as a relative or absolute risk of any excess numbers of childhood carcinomas and hereditary diseases in a population when exposed to radiation, regardless of the radiation dose [11].

Multiple national and international consensus guidelines conclude that there is a negligible risk to the fetus for radiation exposure below 50 mGy. This is further corroborated in the 2008 American College of Radiology (ACR) practice guidelines for imaging pregnant or potentially pregnant patients and supported by the American College of Obstetricians and Gynecologists (ACOG) and the National Council on Radiation Protection and Measurements (NCRPM); fetal radiation doses of less than 50 mGy are not associated with increased fetal anomalies or fetal loss throughout pregnancy [12,13,14,15]. Diagnostic imaging studies commonly used in the initial imaging of trauma patients remain well below this level (Table 1).

Since the fetal radiation exposure of diagnostic imaging studies used in the initial imaging of a pregnant trauma patient remains below the threshold of 50 mGy, this imaging should not be withheld when it is indicated. For diagnostic imaging procedures which do not include direct fetal exposure, such as cranio–cervical, thoracic, or extremity imaging, the fetal radiation exposure remains well below the levels of naturally occurring background radiation during 9 months of pregnancy (0.5–1.0 mGy). A CT of the chest, for instance, provided the fetus is in early gestation and is not included in the field of view of the lower chest/upper abdomen, carries a fetal radiation dose in the range of 0.01–0.06 mGy and is solely related to scatter radiation within the mother [12,13,14].

While the estimated radiation dose for a single-phase post-contrast CT of the abdomen and pelvis with the fetus included in the scan range is higher (estimated at 25 mGy), it is still well below the threshold of 50 mGy. However, if multiple acquisitions through the pelvis are indicated at initial imaging, for example, in patients requiring a CT cystogram or CT angiogram of the aorto–femoral runoff in addition to the standard portal venous phase, radiation exposure to the fetus may exceed the radiation threshold level for deterministic effects. Similarly, for fluoroscopic-guided interventions that would result in exposure to the fetus, for example, for interventional radiology procedures or pelvic stabilization, the 50 mGy threshold can be breached [15,16]. This necessitates careful dose monitoring and utilization of dose reduction methods, including limitation of CT phases to those truly indicated, minimization of fluoroscopy times, using low-pulse-rate fluoroscopy, avoiding magnifications, using collimation and reduced tube current, and avoiding digital subtraction angiography. Finally, using imaging modalities without ionizing radiation for follow-up imaging in pregnant trauma patients, such as ultrasound and MRI, should be considered.

All diagnostic imaging should adhere to the ALARA principle, with justification for any imaging. For pregnant patients who require a larger radiation dose or who may have specific questions regarding their exposure, consultation of a local medical physicist or radiologist can be very helpful. If there is concern about the exposure and dose absorbed by the fetus, an estimated fetal radiation dose can be generated using Monte Carlo simulations [17], allowing the patient or substitute decisions makers to collaborate with their care team for an informed judgement of incurred risk. In their guidelines for imaging during pregnancy, the American College of Obstetrics and Gynecology (ACOG) states that ‘pregnancy termination should not be recommended solely on the basis of exposure to diagnostic radiation’. As in all facets of medical practice, transparent disclosure will lead to collaborative decision making between patient and care team.

## 3. Imaging Modalities for Initial Assessment of the Pregnant Trauma Patient

The initial diagnostic imaging priorities for a pregnant trauma patient in the trauma bay are equivalent to those for a non-pregnant trauma patient. This means that during the initial trauma resuscitation phase, radiographs and FAST (Focused Assessment with Sonography in Trauma, looking for hemoperitoneum and hemopericardium) or e-FAST (extended FAST, which includes assessment for pneumothorax) are performed to guide immediate patient management and indication for CT. MRI and formal diagnostic ultrasound are usually performed once the patient has cleared the initial resuscitation phase.

### 3.1. Radiography

The initial radiographs obtained in a pregnant trauma patient equal those of a non-pregnant trauma patient and usually consist of chest and pelvis radiographs with additional extremity radiographs obtained if injured and time permitting. The chest radiograph can quickly confirm line and tube placement and assess for life-threatening injuries such as pneumothorax, hemothorax, and thoracic osseous injuries. Pelvic radiographs (Figure 1) may identify osseous pelvic injuries that could cause hemodynamic compromise. The presence of pelvic fractures indicates a high likelihood of significant traumatic force to the uterus (and therefore a potential placental and fetal injury), which should prompt swift obstetric assessment, particularly after the time of fetal viability, the gestational age definition of which may vary locally.

### 3.2. Computed Tomography (CT)

Over a 10-year period from 1997 to 2006, Lazarus et al. found that the utilization of radiologic examinations in pregnant women increased by 107%, with the largest per-modality increase in CT [18]. This is in keeping with global trends of increased use of imaging in all patient demographics but, given the potential harmful effects of ionizing radiation, further emphasized the need for justification of the use of CT [19,20]. At our institution, the trauma imaging protocol consists of the following types of CT, depending on clinical indication and justification: a non-contrast CT of the head and cervical spine, a post-contrast arterial phase CT of the chest, and portal venous phase CT of the abdomen and pelvis. A CT angiogram of the head and neck is performed if the patient meets the locally required clinical criteria (the expanded Denver criteria), and extremity run-off CT scans are performed if there is suspicion of vascular injury. For the thoracic and lumbar spine, multi-planar reformations are reconstructed from the torso dataset, and if a CTA of the head and neck was performed, the images of the cervical spine are obtained from that. A delayed urographic phase CT and/or CT cystogram or further additional CT scans are performed on a case-by-case basis, with the decision made at the CT console by a present radiologist or trainee and the trauma service personnel while the patient remains on the CT table. In circumstances when a traumatic injury is confined to a specific body region or an extremity, this team can decide to limit CT interrogation accordingly. Examples of injuries imaged with these protocols can be seen in Figure 2, Figure 3, Figure 4 and Figure 5.

Intravenous and Enteric Contrast

Our trauma CT examinations of pregnant trauma patients do not deviate from non-pregnant patients and are performed with intravenous, iodinated contrast to aid in the detection of vascular and parenchymal injuries. In animal or human studies, intravenous contrast has shown no adverse effects on the fetus. Also, the administration of iodinated contrast material to the mother has not been found to affect fetal thyroid function, and therefore testing of neonatal thyroid function is not necessary [21,22,23].

In the case of penetrating trauma, particularly in the pelvis, oral and/or rectal contrast can be useful in the detection of bowel injuries. Given the importance of assessing bowel wall enhancement for evaluation of its integrity and obscuration of this enhancement by adjacent enteric contrast, at our centre we perform an initial CT with intravenous contrast only followed by one with enteric contrast if deemed necessary. Since this will result in a ‘double dose’ of radiation, providing enteric contrast at the first pass can be considered.

Shielding

There are multiple reasons why the historic use of shielding offered for patient reassurance may potentially be harmful. The improper placement of shielding material can obscure the region of interest and often introduces artifacts, necessitating a repeat study, consequently significantly increasing the radiation dose [24]. When any part of the shieling is situated in the field of view, the high-density shielding material will also affect the automatic exposure control of modern equipment. Machines will automatically increase the tube’s current settings to account for increased density material in the detection zone and in this way increase the fetal dose. Furthermore, scatter radiation originating from the body is prevented from exiting, leading to a further increase in exposure of the fetus. In 2019, the American Association of Physicists in Medicine issued a statement indicating that shielding does not stop that internal scatter. Amongst others, the ACR, the NCRPM, the American Board of Radiologists (ABR), the Society of Pediatric Radiology (SPR), the Canadian Association of Radiologists (CAR), and the main bodies involved in radiation safety and imaging in Europe, in a multidisciplinary European consensus paper, recommend discontinuing the use of shielding [9,10].

### 3.3. Ultrasound

Ultrasound holds many advantages for emergent imaging of the pregnant trauma patient. In addition to its lack of ionizing radiation, it is portable and readily available in the trauma bay. Focused Assessment with Sonography for Trauma (FAST) is a key adjunct of the primary survey performed by the trauma team. FAST was developed to assess for the presence of hemoperitoneum and hemopericardium, with studies demonstrating sensitivities between 85% and 96% and specificities exceeding 98% [25]. Skilled performance and interpretation are paramount; however, even with best efforts, small volumes of fluid (<400 mL) can be difficult to detect [26,27,28]. It is important to remember that the maternal abdominopelvic anatomy alters with increasing duration of gestation and can limit sonographic assessment. Moreover, the low sensitivity of ultrasound for the detection of injuries limits its broader use during resuscitation and it is not a substitute for CT. In pregnant patients, the sensitivity and specificity of ultrasound for detecting intra-abdominal injuries range from 61–83% to 94–100%, respectively [29,30]. In particular, ultrasound poorly demonstrates retroperitoneal hemorrhage and solid organ injury. In addition, it does not sufficiently evaluate the mediastinum, aorta, bowel, or spine, and while it can detect the presence of a pneumothorax, it is time-consuming to assess its volume. Ultrasound assessment of the fetus should be performed as soon as clinically appropriate for assessing the gestational age (particularly as it relates to fetal viability), fetal wellbeing, and placental appearances.

### 3.4. Magnetic Resonance Imaging (MRI)

MRI is not a practical tool for the initial evaluation of traumatic injuries, especially in a hemodynamically unstable patient, regardless of pregnancy status. An MRI examination takes more time and has limitations as to the equipment and peripherals that can be brought in the room, essentially making the environment unsafe with regard to immediate treatment needs. The CAR and ACR recommends against the use of MRI in the initial assessment of the pregnant trauma patient [12,13]. Following stabilization, MRI can have a role in secondary assessment of injuries or as a problem-solving imaging modality, with specific indications such as spinal cord, complex neurological, and soft tissue injuries. MRI may also have a role in reducing radiation exposure in pregnant trauma patients who require follow-up imaging. However, some safety concerns of MRI use in pregnant trauma patients are the heating effects of the radiofrequency pulses used and the effect of acoustic noise on the fetus. Studies have shown no deleterious effects in children who underwent in-utero MRI at 1.5 T up to the age of 9 years [30]. The 2007 American College of Radiology guidance document for safe MRI practices does not differentiate among the pregnancy trimesters and states that all pregnant patients can receive MRI as long as the “risk-benefit ratio to the patient warrants that the study be performed” [31].

Gadolinium-based contrast agents (GBCAs) have limited use for MR imaging in the setting of traumatic injury assessment. GBCAs cross the blood–placental barrier and, depending on fetal gestational age, are excreted by the fetal kidneys into the amniotic fluid and can recirculate. Studies have shown GBCA exposure to pregnant patients is associated with the increased risk of stillbirth and neonatal death in late-term exposure and of rheumatological, inflammatory, and infiltrative conditions in early-term exposure [15]. There is widespread consensus that the use of GBCAs should be avoided in pregnancy where possible. The 2023 ACR Guideline on contrast agents states that these agents should be used with caution with pregnant patients and should “only be used if their usage is considered critical and the potential benefits justify the potential unknown risk to the fetus” [15]. Informed consent for administration of GBCA should be obtained from the patient, whenever possible.

### 3.5. Fluoroscopy

Expansion of procedure types and increased availability of interventional radiology (IR) services globally has changed the blueprint for management of blunt abdominal trauma in all patient cohorts. Given the physiologic stresses associated with operative management and immunologic considerations in the case of splenectomy, non-operative management is prioritized wherever possible and if hemodynamic conditions allow. Angiography with embolization has shown success in the management of most grades of blunt splenic, hepatic, and renal injury. Radiation doses in IR procedures vary depending on the type of the procedure and the area exposed but can quickly exceed radiation levels historically thought to be associated with radiation-induced anomalies [32,33,34]. Adhering to basic principles of time, distance and shielding, aggressive collimation, and use of low-dose fluoroscopy (as opposed to formal angiographic exposures) can all be utilized in order to reduce the dose to both the mother and fetus [35]. These principles can also be more broadly applied when fluoroscopy is used in the operating room for procedures like pelvic or spinal fixation.

## 4. Review of Local Presentations

For the purpose of this review paper, we retrospectively collected all pregnant patients in our institutional trauma registry, which captures all presentations in the trauma bays at our Level-1 Trauma Centre, from September 2002 to June 2023. We documented gestational age at time of injury, mechanism of injury, imaging performed, and maternal and fetal outcome.

During this 21-year period, 76 pregnant patients with major trauma were enlisted in our trauma registry. Of these 76 patients, 45 (59.2%), were in the first trimester (gestational age lest than 13^+6^ weeks, being 13 weeks and 6 days), 15 (19.7%) were in the second trimester (gestational age of week 14 to week 27^+6d^), and 16 (21%) were in the third trimester (gestational age of week 28 to week 40). A wide variety of injury mechanisms were seen, with the highest number of these pregnant patients presenting with motor vehicle collisions (Figure 6). Out of the 76 patients, 11 (14.4%) did not have any CT imaging performed and 3 (3.9%) had imaging deferred until after emergency C-section for fetal distress. A total of 62 (81.5%) patients underwent CT examination. Of those receiving CT, 42/62 (67.7%) had whole-body trauma CT, including CT of the head, CTA of the carotid, and CT of the chest, abdomen, and pelvis, including CT of the whole spine from dedicated reformats. The remaining 20 patients (32.2%) had modified CT imaging, such as either CT of the head and cervical spine or CT of the head and chest. Four out of seventy-six (5.26%) patients had a documented initial FAST in the trauma bay. Forty patients (52.6%) had trauma chest and abdomen radiographs, while twenty-four (31.5%) had only chest radiographs and one (1.31%) patient had only a pelvis radiograph. Nine (11.8%) out of seventy-six patients who did not have any initial trauma radiographs had trauma CTs performed with four pan scans, two CT scans of the chest, abdomen, and pelvis, and three CT scans of the head and cervical spine.

A total of 18/76 (23.6%) cases had a documented discussion regarding the risks of ionizing radiation in the electronic patient chart. There were two (2.63%) documented cases where patients refused to proceed with CT; however, both were subsequently discharged without any major maternal or fetal injuries. Two (2.63%) maternal mortalities were reported in the studied time period; both were drivers in high-speed MVCs and suffered catastrophic head injury with concurrent fetal demise. An additional 10 (13.1%) cases of fetal demise were documented during the admissions of these trauma patients. Out of these, six (60%) were in-hospital spontaneous abortions with two at 8 weeks, three at 6, 7, and 9 weeks, and one at unknown gestational age. Three out of ten (30%) cases of fetal demise were due to placental abruption at 12, 34, and 38 weeks of gestation, while one (10%) case of severe fetal head trauma was seen at 36 weeks of gestation. Maternal Injury Severity Score (ISS) ranged from 1 to 57 in cases of fetal loss.

## 5. Conclusions

Imaging of trauma in pregnancy is a unique and, thankfully, rare occurrence which presents a challenge to all involved, including trauma teams and radiologists. Concern for the potential biological effects of exposure to ionizing radiation of human tissues and, in particular, to that of a developing embryo/fetus can lead to deviation from established imaging protocols that optimally assess for traumatic injuries. However, if indicated by trauma mechanism, clinical parameters, and/or (e-)FAST, obtaining radiographs and subsequent CT should be performed as per usual routine. The rapid and accurate assessment of maternal injuries in order to achieve the quickest maternal stabilization is also the most important goal for good fetal outcome.

Diagnostic-imaging-related radiation exposure during the resuscitative phase in the vast majority of pregnant trauma patients falls below the threshold for significant deleterious effects to the fetus, and the initial diagnostic workup of the pregnant trauma patient should therefore proceed as it would for a non-pregnant trauma patient. If injuries require multiple CT passes through the pregnant uterus, or if intensive fluoroscopic guidance is needed, or short-succession repeat CTs are indicated, consultation of a medical physicist could help guide discussions with the patient with regard to balancing risks and benefits.

## Figures and Tables

**Figure 1 diagnostics-14-00276-f001:**
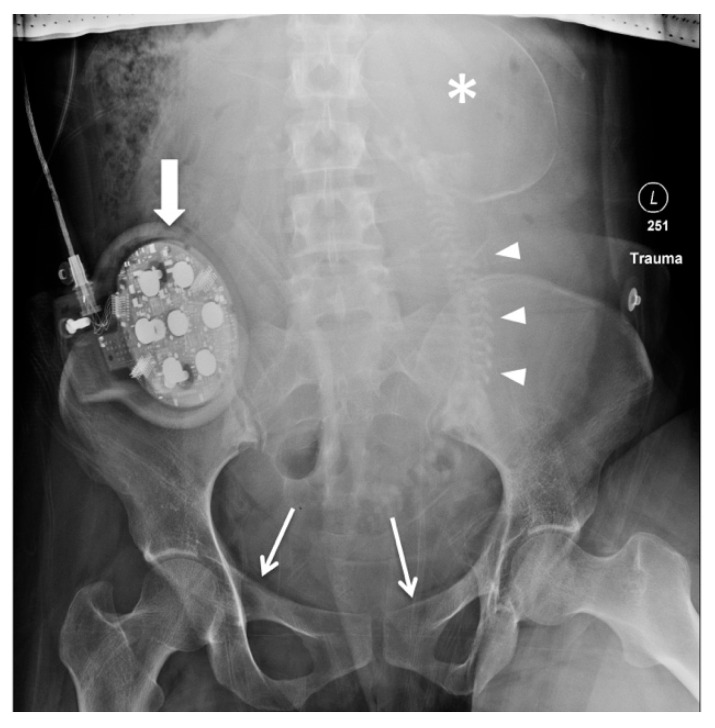
Pelvic radiograph obtained during trauma resuscitation of a pregnant patient at 28 weeks of gestation. Large cranio-caudal field of view was a technical error. The shielding that is present above the level of the fetus at the edge of the image is currently no longer advised. Cardiotocography (CTG) monitor projected over the right iliac wing [block arrow]. Non-displaced fractures of the bilateral pubic rami/pubic root [arrows]. Fetal skeletal structures are visible, such as the skull [asterisk] and spine [arrow heads]. After initial resuscitation demonstrated pelvic stability, patient underwent emergency cesarean section for evidence of fetal compromise, which showed placental abruption.

**Figure 2 diagnostics-14-00276-f002:**
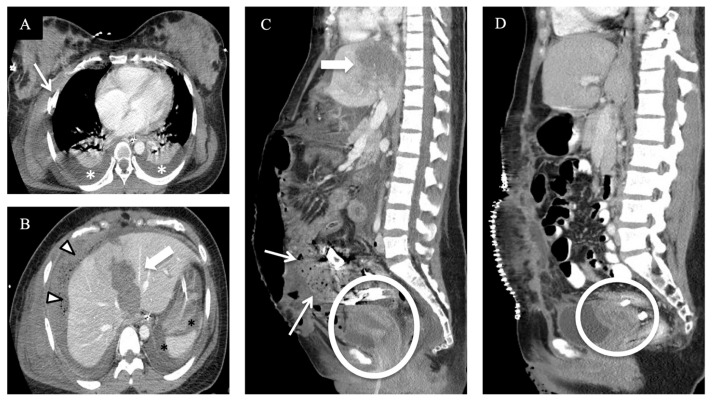
A pregnant patient at 30 weeks of gestation was struck by a car as a pedestrian, presenting hemodynamically very unstable and with a positive FAST, mandating emergent damage control laparotomy prior to imaging. Contrast-enhanced CT axial (**A**,**B**) and sagittal (**C**) images immediately after laparotomy. (**A**) Displaced right rib fracture [arrow] and bilateral hemothorax [asterisks]. (**B**) Grade V liver laceration [block arrow] with perihepatic packing material [arrow heads] and perisplenic hematoma [black asterisks]. (**C**) Grave V liver injury again seen [block arrow] and status after subtotal hysterectomy with remaining part of the uterus [circle] and packing material in pelvis [arrows] with open laparotomy incision. (**D**) Sagittal follow-up CT abdomen and pelvis after removal of the packing material with involuted uterus [circle] and closure of laparotomy incision with enteric contrast in bowel loops.

**Figure 3 diagnostics-14-00276-f003:**
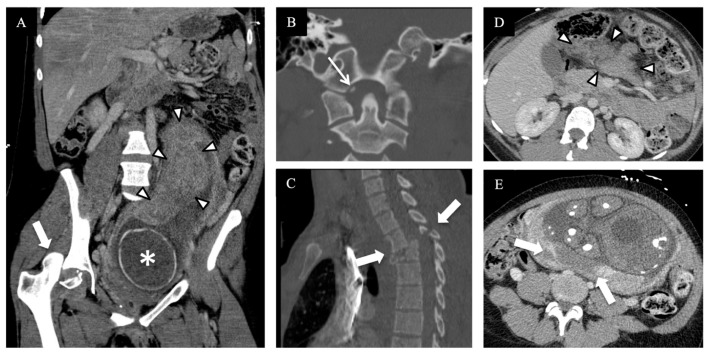
A pregnant patient at 34 weeks of gestation, unbelted when involved in a high-speed MVC, hemodynamically unstable but responsive to resuscitation, with sensory and motor loss below T6 and positive FAST, underwent trauma whole-body CT. (**A**) Oblique coronal contrast-enhanced CT of abdomen and pelvis shows right femoral fracture dislocation [block arrow]. Gravid uterus with fetal skull [asterisk] and placenta [arrow heads]. (**B**) Coronal CT of the cranio–cervical junction spine demonstrating right occipital condyle avulsion fracture [arrow]. (**C**) Sagittal CT of the thoracic spine demonstrating a T3–T4 hyperflexion-distraction injury with translation [block arrows]. (**D**,**E**) Axial contrast-enhanced CT of abdomen and pelvis with a large hematoma in the transverse mesocolon [arrow heads in (**D**)] and placental abruption with a full thickness area of non-enhancement [between block arrows] with fetal structures anterior to it.

**Figure 4 diagnostics-14-00276-f004:**
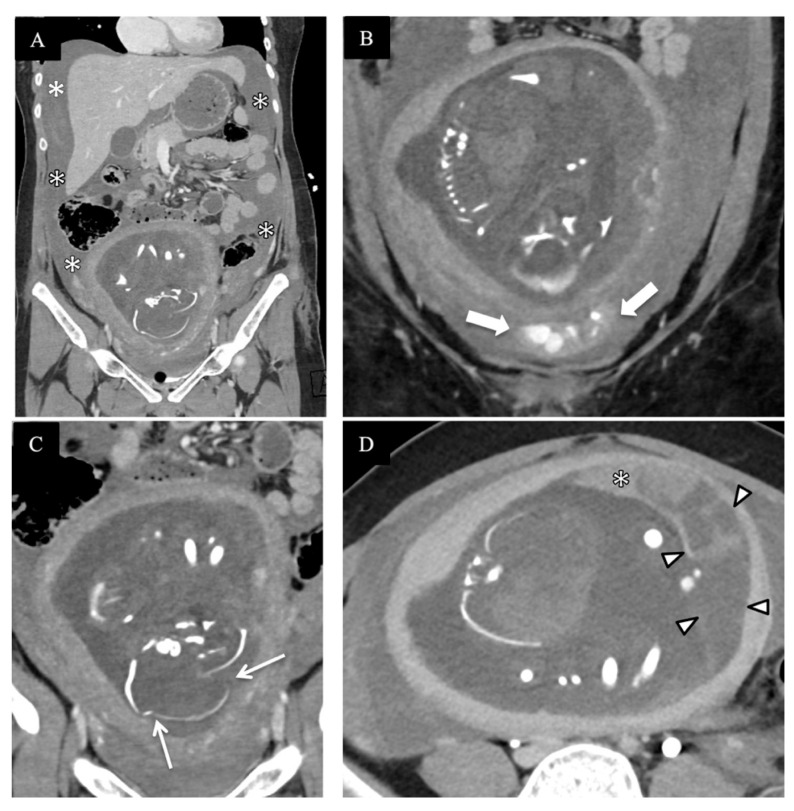
A pregnant patient at 30 weeks of gestation after high-speed MVC, hemodynamically unstable with catastrophic fetal injuries on contrast-enhanced CT of abdomen and pelvis. (**A**) Sagittal image shows large volume hemoperitoneum [asterisks]. (**B**,**C**) Zoomed-in coronal image of the pelvis shows significant contrast extravasation indicating active pelvic hemorrhage [in between block arrows in (**B**)] and fetal skull fractures [arrows in (**C**)]. (**D**) Axial image with heterogeneous placental appearance with focal enhancing component [asterisk] but mostly non- enhancing [between arrowheads], in keeping with placental abruption. Emergent laparotomy confirmed uterine rupture and fetal demise.

**Figure 5 diagnostics-14-00276-f005:**
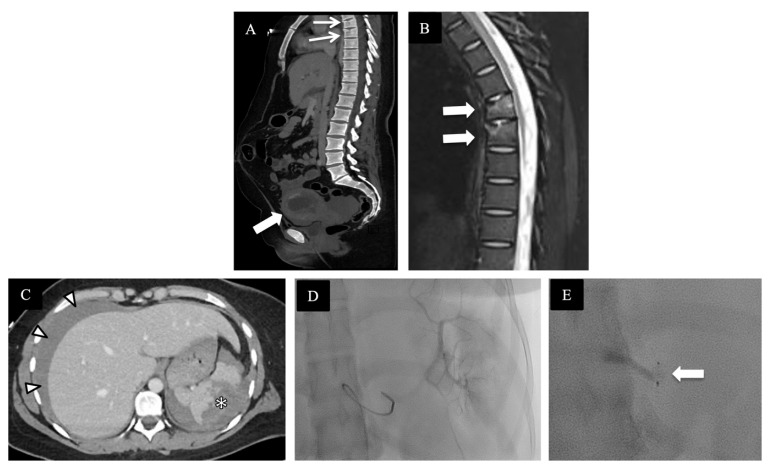
A pregnant patient at 10 weeks of gestation involved in an MVC. (**A**) Sagittal CT of abdomen and pelvis with intrauterine gestation [block arrow] and thoracic vertebral compression fractures [arrows]. (**B**) Sagittal T2-weighted fat-suppressed MR image of the thoracic spine shows T5 and T6 bone marrow edema with compression fractures [block arrows]. (**C**) Axial CT of abdomen displays high-grade splenic injury with perisplenic hematoma [asterisk] and hemoperitoneum [arrow heads]. (**D**,**E**) Procedural fluoroscopic images during splenic artery embolization with a vascular plug placement [block arrow].

**Figure 6 diagnostics-14-00276-f006:**
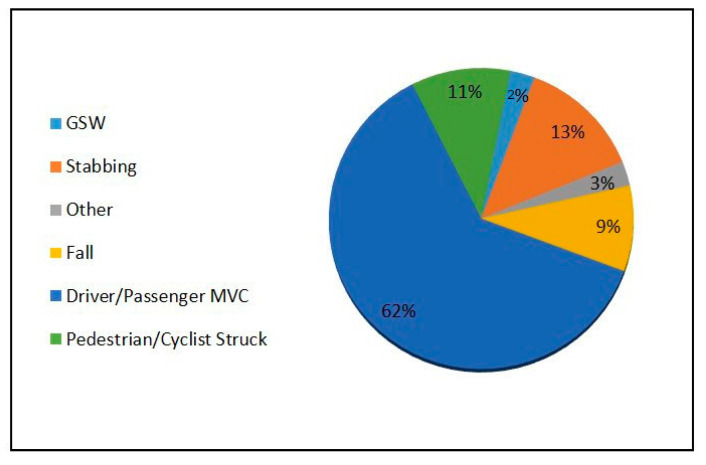
Distribution of mechanisms of trauma in pregnant trauma patients.

**Table 1 diagnostics-14-00276-t001:** Estimated fetal radiation dose for radiographic and CT examinations commonly performed for trauma resuscitation workup. Sourced from references [16].

Examinations Using Ionizing Radiation	Estimated Fetal Dose [mGy]
Radiography	
Cervical spine [AP, lateral]	<0.001
Extremities	<0.001
Chest [PA, lateral]	0.002
Thoracic spine	0.003
Lumbar spine [AP, lateral]	1
Pelvis	1.2
Computed Tomography (CT)	
CT Head	0
CT Chest	0.2
CT Abdomen	4
CT Abdomen and pelvis	25
CT Angiography aorta	34

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
