# Peer review of "Initial Imaging of Pregnant Patients in the Trauma Bay—Discussion and Review of Presentations at a Level-1 Trauma Centre"

_diagnostics, 2024, doi:10.3390/diagnostics14030276_

Round 1
Reviewer 1 Report
Comments and Suggestions for Authors
Although trauma is the leading non-obstetric cause of maternal and fetal mortality and affects an estimated 5-7% of all pregnancies. In fact, most of these are minor trauma, but severe trauma are still rare clinically. First of all, the article used a lot of space to introduce the application of CT in trauma although there are multiple national and international consensus guidelines that there is a negligible risk to the fetus for radiation doses below 50mGy which is safe of trauma patients. This could easily mislead readers and lead to the overuse of CT and radiography in obstetrics. Ultrasound and magnetic resonance imaging (MRI) are more widely used in trauma clinic, but the author mentioned them less and should be added.
Line326-327:1st trimester ( Gestational age less than 13+6weeks; 2nd trimester (week 14 to week 27+6) and 3rd (week28 to week 40).
Comments on the Quality of English LanguageN/A
Author Response
Answers to reviewer comments for manuscript “Diagnostics-2795782”
Reviewer 1:
Comment 1:
Although trauma is the leading non-obstetric cause of maternal and fetal mortality and affects an estimated 5-7% of all pregnancies. In fact, most of these are minor trauma, but severe trauma are still rare clinically. First of all, the article used a lot of space to introduce the application of CT in trauma although there are multiple national and international consensus guidelines that there is a negligible risk to the fetus for radiation doses below 50mGy which is safe of trauma patients. This could easily mislead readers and lead to the overuse of CT and radiography in obstetrics. Ultrasound and magnetic resonance imaging (MRI) are more widely used in trauma clinic, but the author mentioned them less and should be added.
Response authors:
While we value the comment with regards to imaging when patients report to obstetrical or trauma clinics, the manuscript is aimed to describe initial imaging of pregnant patients seen in the trauma bay. In our multidisciplinary experience, this setting can easily lead to deviation from the normal protocol of imaging in trauma bay presentations, which should be avoided. The main imaging modality used in workup of these patients is CT, but there is widespread anxiety of using that in pregnant patients. We firmly believe that we should aim to address this primarily in the current paper. To provide more clarity around this, we have changed the title and have adjusted the wording in the main body where appropriate.
Comment 2:
Line 326-327:1st trimester ( Gestational age less than 13+6weeks; 2nd trimester (week 14 to week 27+6) and 3rd (week28 to week 40).
Response authors:
Thank you for the valuable comment, we have adjusted accordingly.
Reviewer 2 Report
Comments and Suggestions for Authors
This review article is nice written and covers all the important issues related to the principles of imaging for pregnant trauma patients.
Some minor issues regarding phrasing and editing:
1. In page 1, starting from line 7 of the introduction section, 『Using the Injury Severity Score (ISS), a scoring system based on anatomic regions and 38 their regional injury severity with scores ranging from 1 (lowest) to 75 (highest, survivable) [5]. ISS that correlates linearly with mortality, morbidity, hospital stay, and other 40 measures of severity.』. Please make sure these two sentences fit well to the entire paragraph.
2. From the beginning of the last paragraph in page 2, 『There are multiple national and international consensus guidelines that conclude that there is a negligible risk to the fetus for radiation doses below 50mGy, which is well below the exposure from diagnostic imaging studies commonly used in imaging of trauma patients (table 1).』. This meaning of this sentence is not clear, please try to rewrite.
3. The final sentences at the bottom of page 5 are part of figure legends, please adjust font size.
4. First line of page 8, it should be 『the high density….』in stead of 『he high density…』
Author Response
Reviewer 2:
This review article is nice written and covers all the important issues related to the principles of imaging for pregnant trauma patients.
Response authors:
Many thanks for the kind words, which we tremendously appreciate!
Some minor issues regarding phrasing and editing:
- In page 1, starting from line 7 of the introduction section, 『Using the Injury Severity Score (ISS), a scoring system based on anatomic regions and 38 their regional injury severity with scores ranging from 1 (lowest) to 75 (highest, survivable) [5]. ISS that correlates linearly with mortality, morbidity, hospital stay, and other 40 measures of severity.』. Please make sure these two sentences fit well to the entire paragraph.
Response authors:
Thank you for the feedback and opportunity to improve readability, we agree it didn’t flow naturally. We have rewritten the inclusion of the ISS, hopefully with satisfactory result.
- From the beginning of the last paragraph in page 2, 『There are multiple national and international consensus guidelines that conclude that there is a negligible risk to the fetus for radiation doses below 50mGy, which is well below the exposure from diagnostic imaging studies commonly used in imaging of trauma patients (table 1).』. This meaning of this sentence is not clear, please try to rewrite.
Response authors:
Many thanks for the suggestion, we have rearranged the content and rewritten for hopefully better readability.
- The final sentences at the bottom of page 5 are part of figure legends, please adjust font size.
Response authors:
Should be adjusted as suggested now.
- First line of page 8, it should be 『the high density….』in stead of 『he high density…』
Response authors:
Thanks for picking up the error, it is now corrected.
Reviewer 3 Report
Comments and Suggestions for Authors
Dear Authors,
I am glad to have the opportunity to review your work. The aim of the paper was to describe risks and benefits of imaging modalities using ionizing radiation and to provide an overview of imaging findings per imaging modality.
The topic is interesting, although not novel.
The main problem is not defined paper type. The biggest part of the paper is structured as literature review. However, in the Section 4. Review of Local presentations, the authors present statistics from their center, which include original results. This does not qualify as review nor original article and is not in accordance with the journal instructions.
Therefore, I suggest to reject the paper.
Author Response
Reviewer 3:
Dear Authors,
I am glad to have the opportunity to review your work. The aim of the paper was to describe risks and benefits of imaging modalities using ionizing radiation and to provide an overview of imaging findings per imaging modality.
The topic is interesting, although not novel.
The main problem is not defined paper type. The biggest part of the paper is structured as literature review. However, in the Section 4. Review of Local presentations, the authors present statistics from their center, which include original results. This does not qualify as review nor original article and is not in accordance with the journal instructions.
Therefore, I suggest to reject the paper.
Response authors:
Many thanks for the time taken to review our work and for your comments. While we acknowledge that the topic is not novel, we do encounter in practice very regularly that the pregnant trauma patient in the trauma bay causes considerable confusion. Therefore, reiterating the knowledge in this space in current literature in our opinion remains highly valuable. To provide some novelty, we have done the work to review the presentations at our centre and added that to the manuscript. We understand that this adds complexity to categorizing the manuscript, since it does not truly fit either a review type paper or original research. We will leave a decision around whether this is a major issue for the Journal Diagnostics up to the editorial team. If the original data is not desired, it can easily be removed and we are happy to publish that elsewhere. It will decrease the value of this particular paper.
Round 2
Reviewer 1 Report
Comments and Suggestions for Authors
Good job!
I have no more question.
Comments on the Quality of English LanguageN/A
Reviewer 3 Report
Comments and Suggestions for Authors
Dear Authors,
I am glad to have the opportunity to review your work. In the previous round of the review, I have stated that the paper does not qualify as review nor original article and is not in accordance with the journal instructions. The authors in their reply also state that the paper does not truly fit either a review type paper or original research, and they have not done anything to change it. They could have made the changes to reorganize the paper, according to my suggestions, but have rather decided not to do so.
Therefore, I suggest to reject the paper.